# MEMO: Memory-Guided Diffusion for Expressive Talking Video Generation

**Longtao Zheng**[1*], **Yifan Zhang**[2*†], **Hanzhong Guo**[2], **Jiachun Pan**[2], **Zhenxiong Tan**[3],
**Jiahao Lu**[3], **Chuanxin Tang**[2], **Bo An**[1,2], **Shuicheng Yan**[2,3]
[1]*Nanyang Technological University*   [2]*Skywork AI*   [3]*National University of Singapore*
[*]*Equal contribution*   [†]*Project lead*

**Reviewed on OpenReview:** *https://openreview.net/forum?id=uBcHcM7Kzi*

## Abstract

Recent advances in video diffusion models have unlocked new potential for realistic audio-driven talking video generation. However, maintaining long-term identity consistency, achieving seamless lip-audio synchronization, and producing natural, audio-aligned expressions in generated talking videos remain significant challenges. To address these challenges, we propose **M**emory-guided **EMO**tion-aware diffusion (MEMO), an end-to-end audio-driven portrait animation approach to generate identity-consistent and expressive talking videos. Our approach is built around two key modules: (1) a memory-guided temporal module, which enhances long-term identity consistency and motion smoothness by developing causal motion memory to store information from an extended past context to guide temporal modeling; and (2) an emotion-aware audio module, which replaces traditional cross attention with multi-modal attention to enhance audio-video interaction, while detecting emotions from audio to refine facial expressions via emotion-adaptive layer norm. Extensive quantitative and qualitative results demonstrate that MEMO generates more realistic talking videos across diverse image and audio types, outperforming state-of-the-art methods in overall quality, lip-audio synchronization, identity consistency, and expression-audio alignment. Our model and video demos are available at https://memoavatar.github.io.
.

## 1 Introduction

Audio-driven talking video generation [42; 57; 66] has gained significant attention due to its broad impact on areas like virtual avatars and digital content creation, offering transformative possibilities in entertainment, education, and e-commerce. However, compared to text-to-video [17; 45; 43] or image-to-video generation [3], audio-driven talking video generation presents unique challenges. It requires not only generating synchronized lip movements from the audio, but also preserving long-term identity consistency of the reference image and producing natural expressions that align with the emotional tone of the audio. Balancing these demands while ensuring generalization across diverse audio and images makes this task especially challenging.

Recent advances in video diffusion models [57; 65; 9; 12] have enabled more realistic audio-driven talking video generation. Most of these methods autoregressively generate videos with a fixed context window, conditioning on the past 2-4 generated frames. They typically use cross attention to incorporate audio guidance for video generation. Additionally, some incorporate a fixed and predefined emotion label for the whole video to specify the emotion of the generated video [66; 56]. These approaches face three major challenges: long-term identity consistency, lip-audio synchronization, and natural expressions aligned with the audio. First, conditioning on a limited number of motion frames can lead to temporal error accumulation (*i.e.*, drifting), especially when these frames contain artifacts (cf. Figure 4). Second, the use of cross attention limits the level of interaction between audio and video streams. Third, applying a fixed emotion label throughout the video prevents the facial expressions from accurately reflecting the dynamic emotional shifts inherent in the

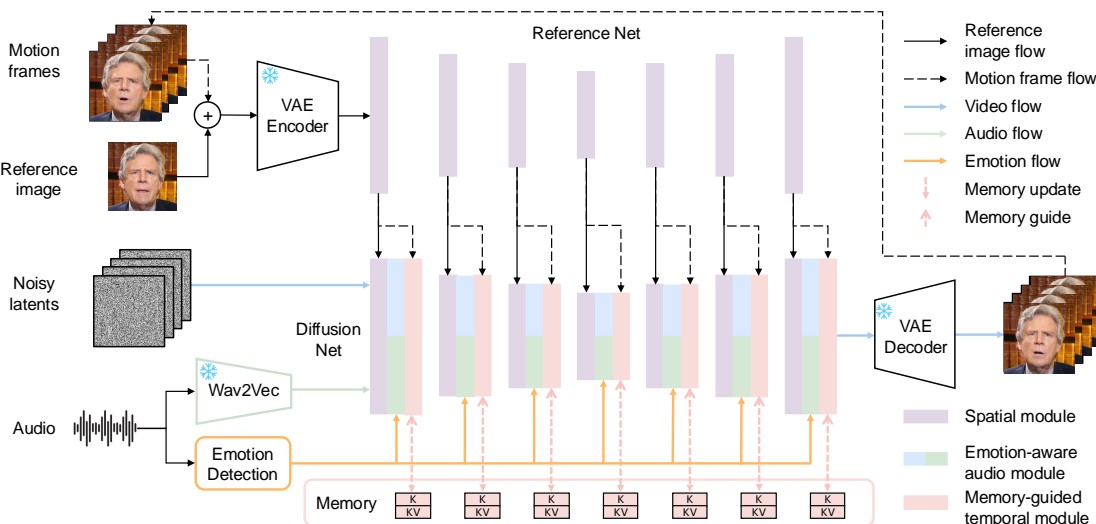

Figure 1: MEMO is structured with a Reference Net and a Diffusion Net. The core innovations of MEMO reside in two key modules within the Diffusion Net: the **memory-guided temporal module** and the **emotion-aware audio module**. These modules work in tandem to deliver enhanced audio-video synchronization, sustained identity consistency, and more natural expression generation.

audio. As a result, existing methods struggle with long-term consistency, lip-audio synchronization, and expression-audio alignment.

In this paper, we propose **M**emory-guided **EMO**tion-aware diffusion (MEMO), an end-to-end audio-driven portrait animation approach. As shown in Figure 1, MEMO builds on two key modules: (1) a memory-guided temporal module and (2) an emotion-aware audio module. To ensure consistent facial identity and smooth transitions across long-duration videos, MEMO develops a memory-guided temporal module (cf. Section 4.1) that maintains a causal motion memory over an extended history of generated frames. This allows the model to use long-term history information to guide temporal modeling through linear attention, resulting in better identity consistency and less error accumulation that may occur in existing diffusion methods (cf. Figure 4). To improve lip-audio synchronization and align facial expressions with audio emotions, MEMO introduces an emotion-aware audio module (cf. Section 4.2). This module replaces audio cross-attention in previous diffusion methods with a more dynamic multi-modal attention mechanism, enabling better interaction between audio and video during the diffusion process. Meanwhile, by dynamically detecting segment-level emotion cues from the audio, this module helps to subtly refine facial expressions via emotion-adaptive layer normalization, enabling the generation of expressive talking videos.

Extensive quantitative results and human evaluations demonstrate that our approach consistently outperforms state-of-the-art methods in overall quality, lip-audio synchronization, expression-audio alignment, identity consistency, and motion smoothness (cf. Table 1 and Figure 9). Additionally, diverse qualitative results highlight MEMO's strong generalization across various types of audio, images, languages, and head poses (cf. Figures 5-8), further showcasing the effectiveness of our method. Lastly, ablation studies further validate the distinct contributions of the memory-guided temporal module, which enhances long-term identity consistency and motion smoothness (cf. Figures 10 and 4), and the emotion-aware audio module, which significantly improves lip-audio alignment and expression naturalness (cf. Figures 13 and 14).

Our method offers two main advantages over previous talking video methods (*e.g.*, EMO [57], Hallo2 [12], and Loopy [25]): (i) MEMO introduces a novel memory-guided temporal module equipped with a causal motion memory mechanism, effectively enhancing long-term temporal consistency and mitigating error accumulation; and (ii) MEMO introduces a new emotion-aware audio module that dynamically detects segment-level audio emotion to guide multi-modal attention, thereby enhancing lip-audio synchronization and expression-audio alignment in talking videos. To our best knowledge, both the causal memory update scheme and the audio emotion awareness mechanism are novel for this field. Our model and code are publicly available on https://memoavatar.github.io to support future research.

## 2 Related Work

Audio-driven talking video generation aims to synthesize realistic and synchronized talking videos given driving audio and a reference face image. Early approaches only focused on learning lip-audio mapping while keeping other facial attributes static [55; 8; 42; 10; 67], which cannot capture comprehensive facial expressions. To resolve this, later research used intermediate motion representations (*e.g.*, landmark coordinates, 3D facial mesh, and 3D morphable models) and decomposed the generation process into two stages, *i.e.*, audio to motion and motion to video [76; 53; 68; 60; 9; 62]. However, they often generate inaccurate intermediate representations from audio, which restricts the expressiveness and realism of videos.

Recent end-to-end methods, like EMO [57] and Hallo [65], can generate vivid portrait videos by fine-tuning pre-trained diffusion models [45]. However, most of these methods used 2-4 motion frames [57; 65; 12] as temporal conditions for autoregressive generation of long videos. However, such a limited frame history can result in error accumulation over time when artifacts appear in the past 2-4 frames. Recently, Loopy [25] increased the number of motion frames to reduce this dependence, and used a temporal segment module to model cross-clip relationships. However, it still uses limited motion frames, whereas our proposed memory-guided linear attention module allows utilizing a broader range of motion frames to provide more comprehensive temporal guidance, thus mitigating error accumulation and enhancing long-term identity consistency. Moreover, unlike prior methods that use cross attention to integrate audio features [57; 65; 12], we introduce multi-modal attention to improve lip-audio synchronization. Additionally, our method dynamically detects fine-grained emotion in the audio, which adaptively guides generation with emotion-adaptive layer normalization, leading to enhanced expression-audio alignment over prior emotion-aware methods [56; 66]. More related studies are provided in Appendix C.

## 3 Problem and Preliminaries

**Problem statement.** Given a reference image and audio as inputs, audio-driven talking video generation [42; 57] aims to output a vivid video that closely aligns with the input audio and authentically replicates real human speech and facial movements. This task is challenging because it requires seamless lip-audio synchronization, realistic head movements, long-term identity consistency, and natural expressions that align with audio. Most existing diffusion-based approaches [57; 65; 9] struggle with issues such as error accumulation, inconsistent identity preservation over time, limited lip-audio synchronization, unnatural expressions, and poor generalization.

**Latent diffusion models and rectified flow loss.** Our method is built upon the Latent Diffusion Model (LDM) [45], a framework designed to efficiently learn generative processes in a lower-dimensional latent space rather than directly operating on pixel space. During training, LDM first employs a pre-trained encoder $\mathcal{E}(\cdot)$ to map high-dimensional images into a compressed latent space, producing latent features $z_0 = \mathcal{E}(I)$. Then, following the principles of Denoising Diffusion Probabilistic Models (DDPM) [22], Gaussian noise $\epsilon$ is progressively added to the latent features over $t$ discrete timesteps, resulting in noisy latent features $z_t = \sqrt{\alpha_t} z_0 + \sqrt{1 - \alpha_t} \epsilon$, where $\alpha_t$ is a variance schedule controlling how much noise is added. The diffusion model is then trained to reverse this noise-adding process by taking the noisy latent representation $z_t$ as input and predicting the added noise $\epsilon$. The objective function for training can be expressed as: $\mathcal{L} = \mathbb{E}_{t, \epsilon \sim \mathcal{N}(0,1)}[\|\epsilon_\theta(z_t | c) - \epsilon\|_2^2]$, where $\epsilon_\theta$ represents the noise prediction made by the diffusion network, and $c$ represents conditioning information such as audio, or motion frames in the context of talking video generation.

Subsequently, Stable Diffusion 3 (SD3) [15] refines this process by incorporating rectified flow loss [32], which modifies the traditional DDPM objective to:

$$\mathcal{L} = \mathbb{E}_{t, \epsilon \sim \mathcal{N}(0,1)}[\lambda(t) \|\epsilon_\theta(z_t | c) - \epsilon\|_2^2], \tag{1}$$

where $\lambda(t) = 1/(1 - t)^2$ and $z_t$ is reparameterized using linear combination as $z_t = (1 - t) z_0 + t\epsilon$. This formulation leads to both better training stability and more efficient inference. In light of these advantages, we adopt the rectified flow loss from SD3 in our training.

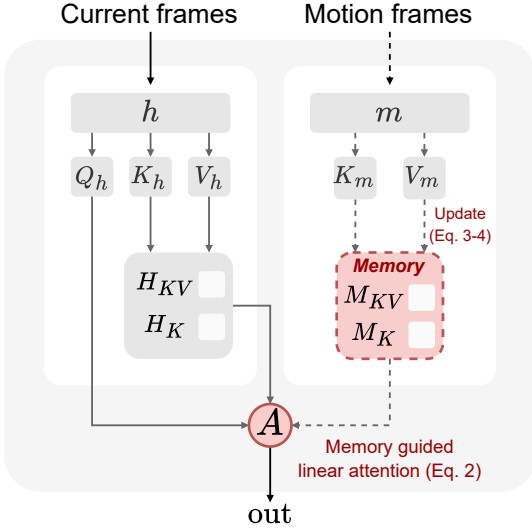
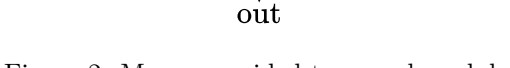

Figure 2: Memory-guided temporal module.

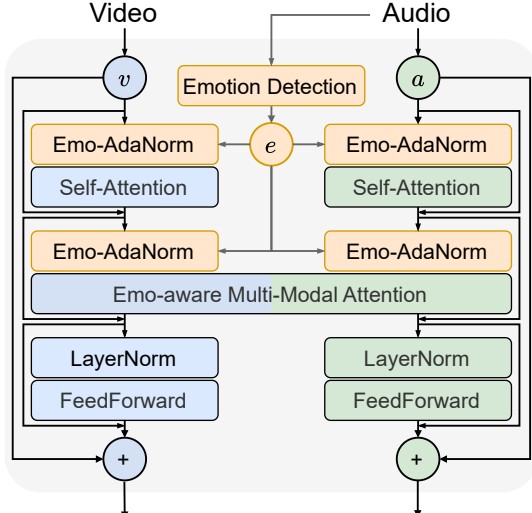

Figure 3: Emotion-aware audio module.

## 4 Method

As illustrated in Figure 1, MEMO is an end-to-end audio-driven diffusion model for generating identity-consistent and expressive talking videos. Similar to previous diffusion-based approaches [57; 65], MEMO is built around two main components: a Reference Net and a Diffusion Net. The main contributions of MEMO lie in two key modules within the Diffusion Net: the **memory-guided temporal module** (cf. Section 4.1), and the **emotion-aware audio module** (cf. Section 4.2), which work together to achieve superior audio-video synchronization, long-term identity consistency, and natural expression generation. Moreover, MEMO introduces a decomposed training strategy (cf. Section 4.3), along with a new data processing pipeline (cf. Section 4.4).

### 4.1 Memory-Guided Temporal Module

Most existing diffusion methods [57; 65; 9] generate talking videos in an autoregressive manner by dividing the audio into clips of 12-16 frames and using the past 2-4 generated frames to guide the generation of the next video clip. Although this strategy can model short-term dependencies, it often struggles with maintaining long-term consistency. If artifacts are generated in the 2-4 conditioned motion frames, these errors tend to accumulate as generation progresses, leading to visual distortions that degrade both video quality and identity consistency (*e.g.*, Hallo2 [12] in Figure 4). To address these issues, we propose a memory-guided temporal module for long-term consistency. This module consists of two key designs: (1) memory-guided linear attention, and (2) causal memory update.

**Memory-guided linear attention.** To alleviate temporal error accumulation, our main intuition is that leveraging a longer motion history can provide richer temporal context than solely relying on the most recent 2-4 frames. Inspired by this, we propose to increase the number of motion frames to better guide temporal modeling. However, previous approaches use self-attention [57; 25] to model temporal relationships between frames, which requires storing all key-value pairs. As the number of motion frames increases, GPU memory overhead also increases. Therefore, it is challenging for self-attention to effectively utilize the history beyond its limited-length context window. To address this issue, we replace self-attention with linear attention [27] for temporal modeling. Specifically, denoting query as $Q$, key as $K$, and value as $V$, the output of linear attention for $i$-th frame is: $\text{out}_i = \phi(Q_i)^\top \big( \sum_{j=1}^{f} \phi(K_j) V_j^\top \big) / \phi(Q_i)^\top \sum_{j=1}^{f} \phi(K_j)$, where $f$ is the number of frames and $\phi$ is the activation function (we use softmax in this work).

To incorporate more motion frames to guide generation, we construct a motion memory to record the past generated frames. As shown in Figure 2, let the latent features of motion frames be $m \in \mathbb{R}^{f \times d}$ and the latent features of current frames be $h \in \mathbb{R}^{f \times d}$, where $d$ is the dimension of latent features. Linear attention processes these latent features via learnable matrices, which transform them into queries ($Q_h$), keys ($K_h, K_m$), and values ($V_h, V_m$). We then define the motion memory for the past $f$ frames as two matrices: $M_{KV}^f = \sum_{i=1}^{f} \phi(K_{m,i})V_{m,i}^\top$ and $M_K^f = \sum_{i=1}^{f} \phi(K_{m,i})$, which occupy constant GPU memory irrespective of $f$. Based on the memory, the output of the memory-guided linear attention is formulated as:

$$\text{out} = \frac{\phi(Q_h)^\top (H_{KV} + M_{KV})}{\phi(Q_h)^\top (H_K + M_K)},\tag{2}$$

where $H_{KV} = \phi(K_h)V_h^\top$ and $H_K = \phi(K_h)$.

**Causal memory update.** After each generation of $f$ frames, we update the memory $M^f$ by incorporating information from these newly generated frames. In formal, the memory update when adding the latest $a$ frames to the memory with $b$ motion frames is:

$$M_{KV}^{a+b} \leftarrow \gamma^a M_{KV}^b + \sum_{j=1}^{a} \gamma^j \phi(K_{h,j})V_{h,j}^\top,\tag{3}$$

$$M_K^{a+b} \leftarrow \gamma^a M_K^b + \sum_{j=1}^{a} \gamma^j \phi(K_{h,j}).\tag{4}$$

Here, $\gamma$ is a frame-level decay factor that modulates the influence of motion frames, with more recent frames exerting greater impact, reflected through the exponentiation by $i$. Such a decay scheme is critical, as a unified positional encoding across different video subsegments is infeasible. Instead, we use causal memory decay to provide implicit positional encoding, which enables more effective memory updates to capture long-term dependencies.

## 4.2 Emotion-Aware Audio Module

Previous talking video methods face two key limitations in handling audio conditions. First, the cross-attention commonly used in prior approaches [57; 65; 9] provides limited interaction between audio and video modalities, restricting their effectiveness in achieving better lip-audio synchronization. Second, existing emotion-aware techniques [66; 56] typically rely on a fixed emotion label for the entire video, which fails to capture the dynamic emotional shifts in audio, causing misalignment between the generated facial expressions and the audio's emotion. To address these challenges, we propose a new emotion-aware audio module to improve lip-audio synchronization and align facial expressions with the underlying audio emotion. Specifically, our audio module is distinguished by the following three core designs: (1) multi-modal attention, (2) audio emotion-aware diffusion, and (3) emotion decoupling training.

**Multi-modal attention.** Our emotion-aware audio module replaces the traditional cross attention with a more dynamic multi-modal attention mechanism. Specifically, cross attention aligns video and audio by conditioning the process of video features $v$ on audio features $a$, which can be formalized as minimizing the loss function $\mathcal{L}_{\theta_{(v|a)}} = \mathbb{E}_{t,\epsilon\sim\mathcal{N}(0,I)}[\lambda(t)\|\epsilon_\theta(v_t|a)-\epsilon\|_2^2]$. In contrast, we explore multi-modal attention, which jointly processes both video and audio inputs by minimizing the loss function $\mathcal{L}_{\theta_{(v,a)}} = \mathbb{E}_{t,\epsilon\sim\mathcal{N}(0,I)}[\lambda(t)\|\epsilon_\theta(v_t,a) - \epsilon\|_2^2]$. Specifically, as shown in Figure 3, we use two modality-specific branches for audio and video, within which an intermediate step concatenates their outputs into a unified token sequence for multi-modal self-attention. This enables better video-audio interaction during the diffusion process, thus achieving enhanced lip-audio synchronization.

**Audio emotion-aware diffusion.** To enhance expression-audio alignment, our audio module seeks to dynamically detect audio emotions to guide audio-video interaction. To this end, we train a new emotion detector to extract emotion from audio (See Appendix E for more implementation details and robustness discussions of the audio emotion detector), recognizing eight distinct emotions: `angry`, `disgusted`, `fearful`, `happy`, `neutral`, `sad`, `surprised`, and `others`. To improve the robustness of emotion labels, we conduct detection at the segment level. Each segment's emotion is determined by the most frequently detected emotion across all its frames, where each frame's emotion is evaluated using audio features from a 3-second sliding window centered on that frame.

We then incorporate the segment-level audio emotion via emotion-adaptive layer normalization (Emo-AdaNorm) for expression-audio alignment. As shown in Figure 3, the detected emotion of each segment is first projected into an emotion embedding $e$, which is then integrated into each layer via Emo-AdaNorm to guide multi-modal attention. Specifically, $e$ is processed through a linear modulation [40] to produce a scaling factor $\alpha$ and a shifting factor $\beta$. These factors adaptively modulate the LayerNorm (LN) outputs, i.e., Emo-AdaNorm$(x) = (1+\alpha)$LN$(x)+\beta$, thereby conditioning the normalization explicitly on the detected emotion labels. This approach transforms Eq. 1 into the following emotion-conditioned flow loss:

$$\mathcal{L}_{\theta_{(v,a|e)}} = \mathbb{E}_{t,\epsilon\sim\mathcal{N}(0,I)}[\lambda(t)\|\epsilon_\theta(v_t,a|e) - \epsilon\|_2^2]. \tag{5}$$

During inference, we use classifier-free guidance (CFG) [21] to control the effect of emotion on the output. The emotion-aware output is computed as

$$\tilde{\epsilon}_\theta(v_t,a|e) = (1+w)\epsilon_\theta(v_t,a|e) - w\epsilon_\theta(v_t,a), \tag{6}$$

where $w$ is the CFG scale to control the influence of the emotion condition. Eq. 6 is simplified to highlight CFG for emotion, but in practice, CFG is similarly applied to both audio and the reference image. It should be noted that the overall emotional tone of the generated talking video is largely inferred from the facial expression in the reference image. The audio emotion serves more as a subtle adjustment to enhance the emotional expression in response to audio cues, rather than completely overriding the facial expression provided by the reference image.

**Emotion decoupling training.** To improve the effect of audio emotion in talking videos, we develop an emotion decoupling training strategy to separate the expression in the reference image from the audio emotion. Specifically, for training video clips sourced from MEAD [61]—which provides both speaker identity and emotion labels—we avoid using a reference image from the same video clip. Instead, we randomly select a reference image of the same person but with a different emotion. This encourages a better disentanglement between the reference image's expression and the audio-induced emotion, enabling our emotion-aware module to better refine facial expressions in alignment with the audio. Our method also supports replacing the detected audio emotion label with a manually specified emotion label, if desired.

## 4.3 Training Strategy Decomposition

The training of MEMO is divided into two progressive stages, each with specific objectives.

**Stage 1: Face domain adaptation.** We initialize Reference Net and the spatial module of Diffusion Net with the weights of SD 1.5 [45]. In this stage, we adapt Reference Net, the spatial attention modules of Diffusion Net, and the original text cross-attention module to the face domain with the rectified flow loss (cf. Eq. 1), ensuring these components capture facial features effectively.

**Stage 2: Emotion-decoupled robust training.** We then integrate the emotion-aware audio module and memory-guided temporal module into the Diffusion Net. Initially, we perform a warm-up training phase for the newly added modules, keeping the modules in Stage 1 fixed. After the warm-up, all modules are jointly trained. In this stage, we use the emotion-conditioned flow loss (cf. Eq. 5) and scale up the dataset to include all processed data for more comprehensive training. Here, we adopt the emotion decoupling training strategy (cf. Section 4.2) only when the training video clips are sourced from MEAD. Moreover, we found that some noisy data persisted even after applying our data processing pipeline (cf. Section 4.4), making diffusion training unstable and leading to biased model optimization. To mitigate this, we further develop a robust training strategy that filters out data points with loss values suddenly exceeding a specific large value (0.1 in our case), as the emotion-conditioned flow loss in our method typically converges and fluctuates around 0.03. Please refer to Appendix D for more implementation details.

## 4.4 Data Processing Pipeline

We collect a comprehensive set of open-source datasets, such as HDTF [73], VFHQ [63], CelebV-HQ [77], MultiTalk [54], and MEAD [61], along with additional data collected by ourselves. The total duration of

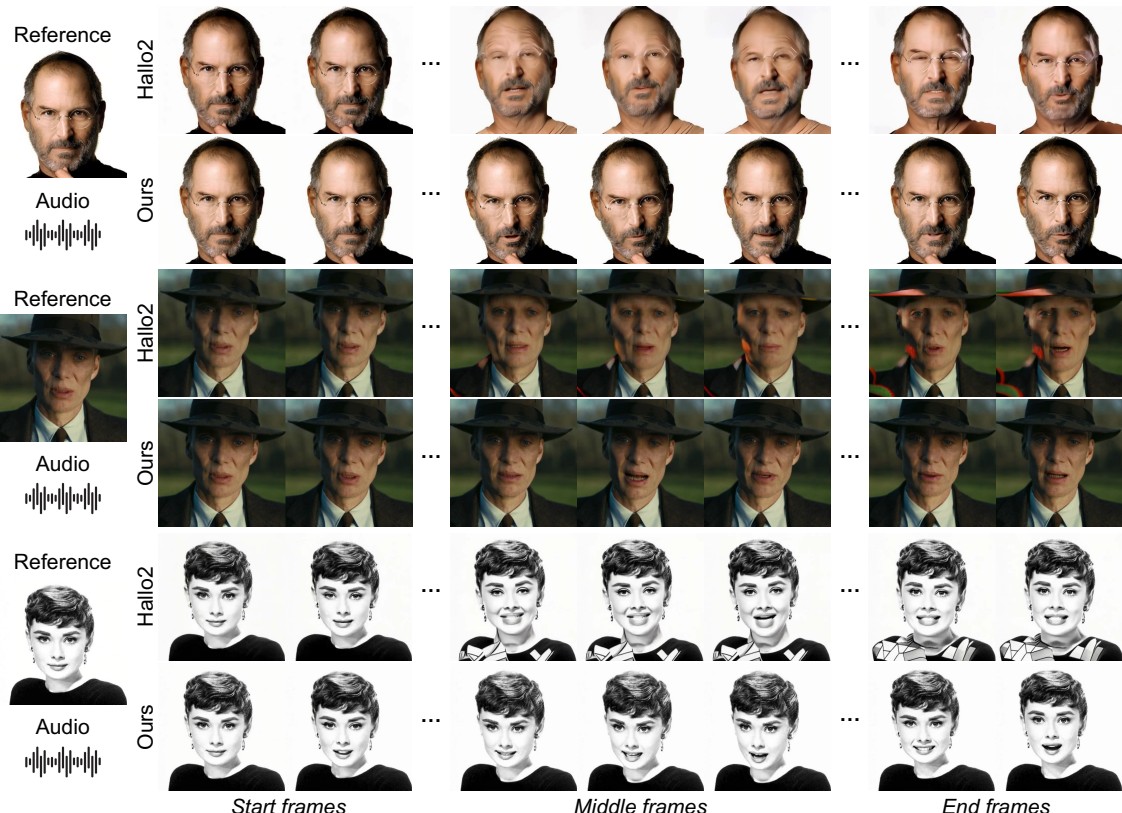

Figure 4: MEMO generates long talking videos with improved identity consistency, lip-audio alignment, and motion smoothness. In contrast, existing methods (*e.g.*, Hallo2 [12]) are prone to drifting during autoregressive generation, especially when the last 2-4 generated frames used as temporal conditions contain artifacts, leading to inconsistent identity. Please refer to the supplementary material for video demos.

these raw videos exceeds 2,200 hours. However, as illustrated in Appendix F, we find that the overall quality of the data is poor, with numerous issues such as lip-audio misalignment, missing heads, multiple heads, occluded faces by subtitles, extremely small face regions, and low resolution. Directly using these data for model training results in unstable training, poor convergence, and terrible generation quality.

To further obtain high-quality talking head data, we developed a dedicated data processing pipeline for talking head generation. The pipeline consists of five steps: First, we perform scene transition detection and trim video clips to a length of less than 30 seconds. Second, we apply face detection, filtering out videos with no faces, partial faces, or multiple heads, and use the resulting bounding boxes to extract talking heads. Third, we use an Image Quality Assessment model [51] to filter out low-quality and low-resolution videos. Fourth, we apply SyncNet [42] to remove videos with lip-audio synchronization issues. Lastly, we manually assess the lip-audio synchronization and overall video quality for a subset of the data to ensure more accurate filtering. After completing the entire pipeline, the total duration of our processed high-quality videos is about 660 hours.

# 5 Experiments

## 5.1 Experimental Setup

**Evaluation benchmarks.** We create two out-of-distribution (OOD) datasets to evaluate MEMO's performance and generalization. For the first OOD dataset, we sample 150 video clips from the VoxCeleb2 [37]

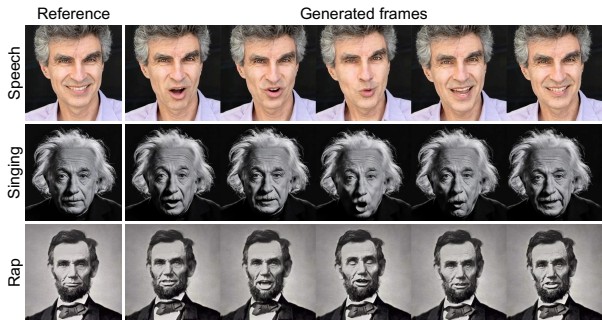
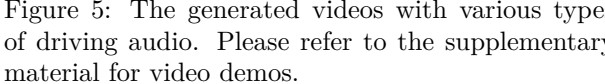
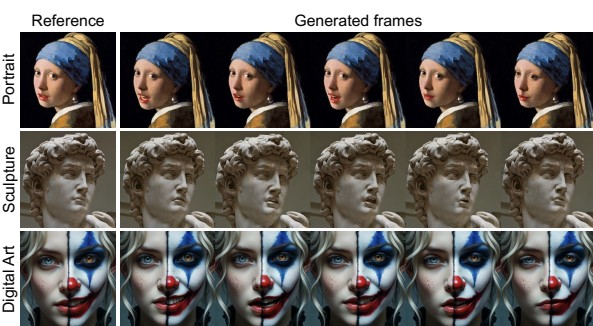

Figure 5: The generated videos with various types of driving audio. Please refer to the supplementary material for video demos.

Figure 6: The generated videos with various types of reference images. Please refer to the supplementary material for video demos.

test set, with videos of various celebrities. Similarly, we create the second OOD dataset with 150 clips across a more diverse set of audio, backgrounds, ages, genders, and languages.

**Evaluation metrics.** We adopt a suite of metrics to evaluate the overall quality and lip-audio synchronization of the generated videos. FVD [58] measures overall video quality with the distance between the distributions of real and generated videos. FID [20] evaluates the quality of individual frames by comparing feature distributions extracted from a pre-trained model. For lip-audio synchronization, we report Sync-C and Sync-D [11] using a pre-trained discriminator model [42]. To evaluate identity consistency, we compute the cosine similarity between face embeddings (Identity). To assess facial expressiveness, we use Expression-FID (E-FID), following [57].

**Baselines.** We compare our method against several state-of-the-art baselines with publicly available model checkpoints. The baselines include both two-stage methods with intermediate representations and end-to-end diffusion methods. V-Express [60], AniPortrait [62] and EchoMimic [9] are two-stage methods using intermediate representations like landmarks, while Hallo [65] and Hallo2 [12] are recent end-to-end diffusion-based models.

**Role of training data vs. architecture.** We acknowledge that stronger training data can partially contribute to overall gains. MEMO is trained on our curated dataset, while compared baselines are evaluated using officially released checkpoints, each trained with its own (often undisclosed) data. As most baselines do not release complete training data and code, fully controlled retraining under one unified pipeline is infeasible. To reduce this confounder, all module-wise ablations in Tables 2 and 3 are trained and evaluated with the same setting to isolate the contributions of the proposed memory and emotion modules.

## 5.2 Quantitative Results

**Performance on two OOD test sets.** Table 1 reports the quantitative results on two OOD test sets. Our method consistently outperforms all baselines in terms of FVD, FID, and Sync-D metrics, indicating better video quality and lip-audio synchronization. These results also demonstrate the improved generalization abilities of MEMO to unseen identities and audio. In addition to achieving the best or comparable results on Sync-C and E-FID, MEMO shows clear advantages in identity consistency across two test sets.

**Human evaluation.** To better benchmark the quality of generated talking videos, we conduct human studies based on five subjective metrics in several challenging scenarios, *e.g.*, singing, rap, and multi-lingual talking video generation. Specifically, our analyses are based on the overall quality, motion smoothness, expression-audio alignment, lip-audio synchronization, and identity consistency. As shown in Figure 9, our method achieves the highest scores across all criteria in human evaluations, much higher than compared methods. This further demonstrates the effectiveness of our approach.

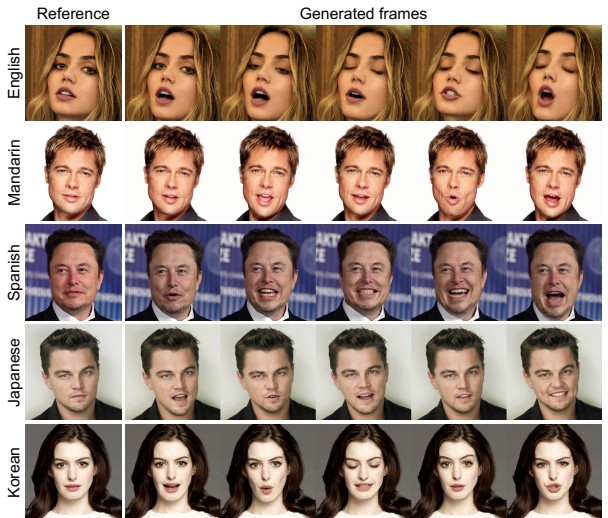

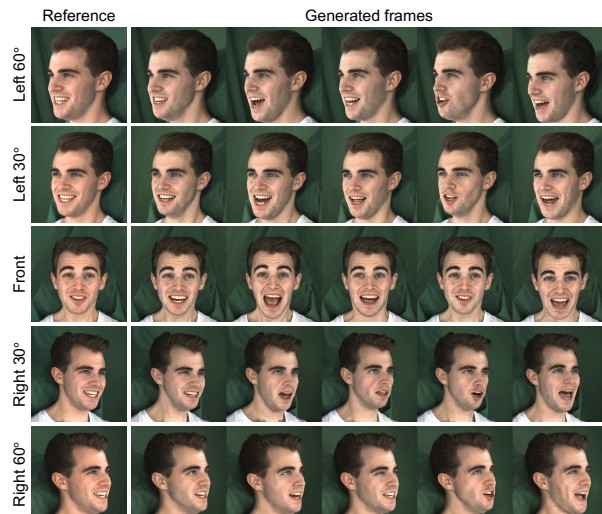

Figure 7: The generated videos on driving audio with different languages. Please refer to the supplementary material for video demos.

Figure 8: The generated videos with reference images of different head poses. Please refer to the supplementary material for video demos.

Table 1: Quantitative results of video quality, lip-audio synchronization, identity consistency, and expressiveness on two OOD test datasets. MEMO consistently outperforms existing talking video baselines. For E-FID, "-" denotes omission where the face-recognition model cannot reliably identify enough faces to yield a statistically reliable estimate.

| Method | VoxCeleb2 test set | | | | | | Collected OOD dataset | | | | | |
|---|---|---|---|---|---|---|---|---|---|---|---|---|
| | FVD↓ | FID↓ | Sync-C↑ | Sync-D↓ | Identity↑ | E-FID↓ | FVD↓ | FID↓ | Sync-C↑ | Sync-D↓ | Identity↑ | E-FID↓ |
| SadTalker [68] | 397.0 | 71.7 | 5.5 | 8.6 | 0.66 | 10.4 | 288.7 | 48.3 | 5.3 | 10.6 | 0.83 | - |
| AniPortrait [62] | 333.2 | 45.5 | 2.8 | 11.0 | 0.80 | 9.5 | 238.7 | 31.2 | 3.5 | 10.5 | 0.90 | **6.3** |
| V-Express [60] | 418.9 | 58.9 | 6.4 | 8.2 | 0.82 | - | 315.2 | 46.7 | **6.1** | 9.5 | 0.87 | - |
| Hallo [65] | 330.4 | 41.6 | 6.6 | 8.0 | 0.75 | 7.0 | 231.1 | 31.9 | 5.4 | 9.3 | 0.88 | 7.3 |
| Hallo2 [12] | 302.0 | 41.6 | 6.4 | 8.0 | 0.77 | 7.3 | 223.1 | 29.8 | 5.5 | 9.3 | 0.88 | 6.9 |
| EchoMimic [9] | 293.9 | 43.8 | 3.7 | 10.1 | 0.67 | - | 223.9 | 39.9 | 4.5 | 9.8 | 0.89 | 6.9 |
| **MEMO (Ours)** | **254.3** | **31.7** | **6.8** | **7.4** | **0.85** | **6.2** | **161.1** | **24.9** | 5.7 | **9.2** | **0.92** | 6.4 |

## 5.3 Qualitative Results

**Long-duration talking video generation.** Figure 4 and Figure 11 demonstrate that MEMO can generate long-duration videos while consistently maintaining the subject's facial features and expression fidelity over thousands of frames. The resulting videos exhibit smooth motion and high temporal coherence, showcasing the robustness of our approach for long video synthesis. These capabilities make MEMO a superior choice over existing methods for applications requiring extended video content with stable generation quality.

**Generalization to different types of audio.** We evaluate MEMO on various audio types, including speeches, songs, and raps. In Figure 5, MEMO consistently generates synchronized lip movements across diverse audio types. It performs well with both expressive songs, which demand nuanced emotional alignment, and raps, which require rapid lip-audio synchronization. This verifies MEMO's strong generalization to various types of driving audio.

**Generalization to different styles of reference images.** Figure 6 shows MEMO's performance on challenging reference images of diverse styles, such as portraits, sculpture, and digital art images. Despite these styles deviating significantly from our training data, MEMO maintains robust generation quality without producing noticeable artifacts, demonstrating its generalizability to OOD images.

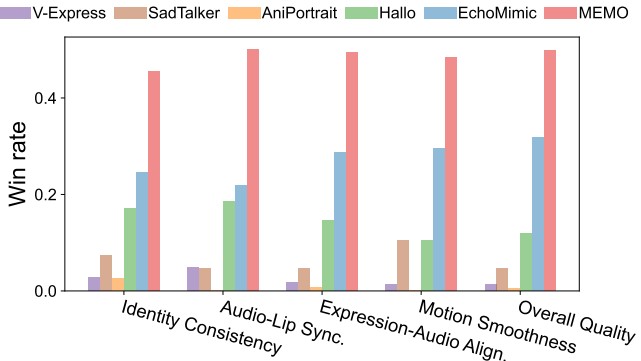
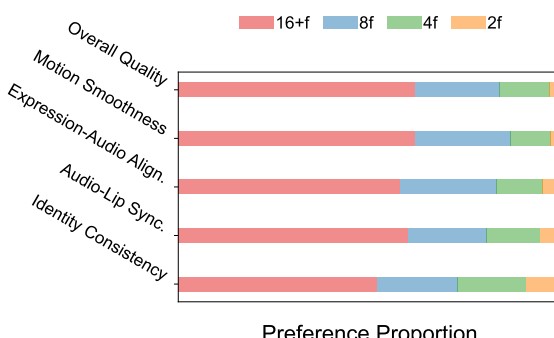

Figure 9: Human preferences among MEMO and baselines in terms of five metrics.

Figure 10: Human preferences on the number of motion frames ($f$), where $16+f$ indicates a context beyond 16 frames.

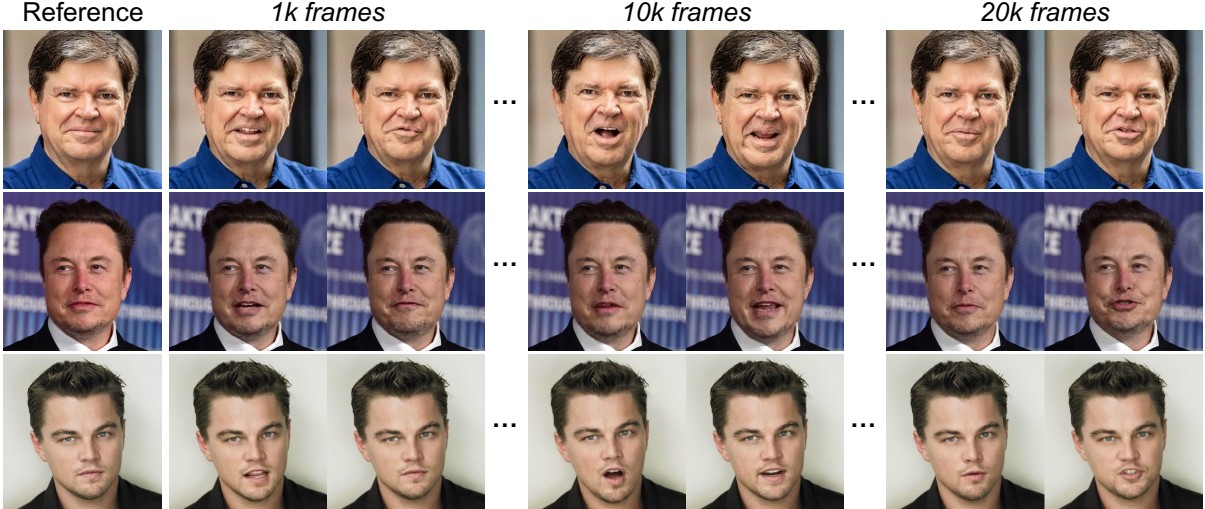

Figure 11: MEMO can generate long-duration videos with alleviated error accumulation and maintain identity consistency. Please refer to the supplementary for video demos.

**Generalization to multilingual audio.** Figure 7 demonstrates that MEMO achieves robust generalization across multilingual audio inputs, such as English, Chinese, Spanish, Japanese, and Korean. Despite most of our training data being in English, our approach effectively generates lip movements synchronized with the given multilingual audio while capturing rich, realistic facial expressions.

**Generalization to different head poses.** Figure 8 shows the generated videos of MEMO with reference images of varying head poses, including frontal views and multiple side angles. This demonstrates that our method can generate realistic talking videos across different angles while maintaining consistency in facial appearance and expression. In Figure 6, we also demonstrate that MEMO generalizes to different head poses for more challenging reference images. While enhancing head movements is not the focus of our work, our method can still generate talking videos with better head motion diversity compared to previous open-source approaches (cf. Figure 12). This increased motion diversity enhances the naturalness and expressiveness of the generated videos.

## 5.4 Ablation Studies

To evaluate the effects of method components, we conduct ablation studies based on quantitative metrics, human evaluation, and qualitative analysis. More discussions are provided in Appendices A and B.

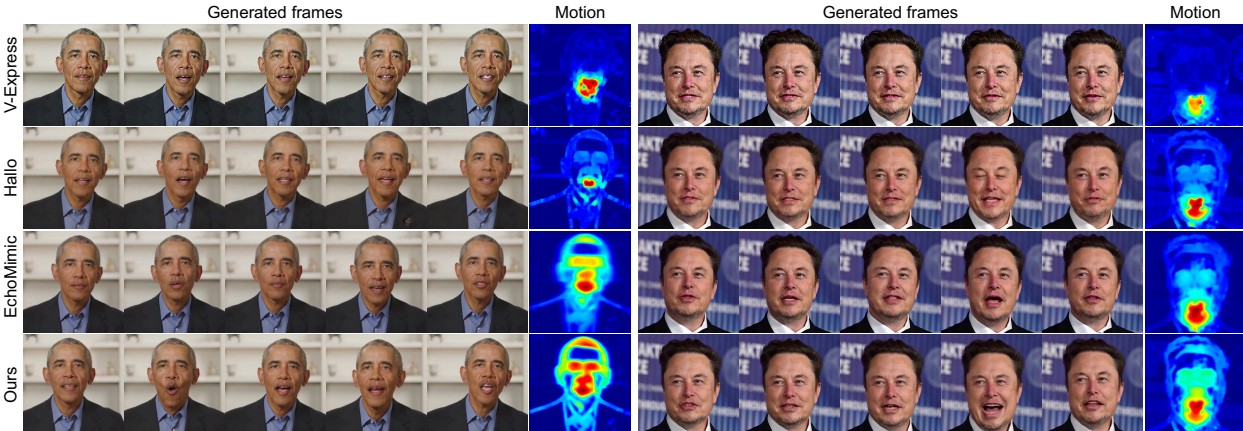

Figure 12: MEMO can generate talking videos featuring a wider range of smooth head movements and more emotional facial expressions, illustrated in both visualization and heatmaps.

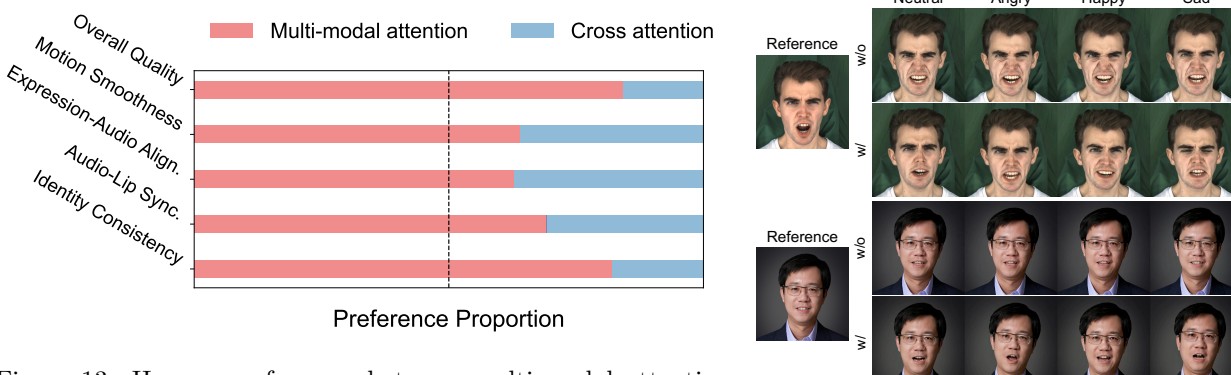

Figure 13: Human preferences between multi-modal attention and cross attention for integrating audio conditions in the audio module.

Figure 14: Ablation of emotion decoupled training.

**Effects of memory module.** We evaluate the impacts of our memory module through both human evaluation and quantitative ablations. In Figure 10, longer memory significantly improves temporal coherence, overall quality, motion smoothness, identity consistency, and lip-audio alignment, while short motion frames lead to worse performance. This demonstrates the effectiveness of our memory-guided temporal module and also explains why our method can alleviate temporal error accumulation in Figure 4, whereas Hallo2 is prone to error accumulation. For quantitative ablations, we control the history information available to the memory module and compare three settings: memory with only the initial reference image, memory with only the most recently generated frames, and full memory. As shown in Table 2, using the full memory yields the best FVD, Sync-C, and head-motion statistics.

**Effects of multi-modal attention.** We further investigate the impact of the multi-modal attention through human evaluations. Results in Figure 13 underscore the effectiveness of multi-modal attention over cross attention in terms of the overall video quality and lip-audio alignments.

**Effects of emotion-aware scheme.** To investigate the effects of our emotion-aware scheme (*i.e.*, emotion-aware diffusion and emotion decoupling training), we conduct ablations through both quantitative metrics and qualitative analysis. In Figure 14, using the same reference image and different emotion labels, we compare the frames generated by our method with and without emotion decoupling training at the same audio moment. The results demonstrate that our emotion-aware module with emotion decoupling training can effectively refine facial expressions to align with the specified emotion labels. This finding suggests that,

Table 2: Controlled quantitative ablation of the memory module on the collected OOD dataset.

| Method | FVD↓ | Sync-C↑ | Head Motion↑ |
|---|---|---|---|
| Reference image only | 175.2 | 4.32 | 1.54 |
| Recent frames only | 165.9 | 5.67 | 4.03 |
| Full memory (Ours) | **161.1** | **5.74** | **4.19** |

Table 3: Controlled quantitative ablation of the emotion module on the collected OOD dataset.

| Method | FVD↓ | FID↓ | LPIPS↓ |
|---|---|---|---|
| No emotion | 163.0 | 25.0 | 0.268 |
| Ours | **161.1** | **24.9** | **0.263** |

with our detected emotion labels, MEMO can generate facial expressions that match the audio emotion. Moreover, this comparison also validates the necessity of our emotion decoupling training. Quantitatively, we manually control the emotion label to static to test visual quality metrics to validate the effectiveness of the emotion module. As shown in Table 3, the emotion module consistently improves FVD, FID, and LPIPS over the variant without emotion conditioning. While these improvements are modest in automatic metrics, they are consistent with the stronger user preference observed in our human study, suggesting that the emotion-aware design improves both perceptual expressiveness and overall generation quality. We do not report off-the-shelf emotion-classifier scores because, in our experiments, such classifiers are unreliable on synthesized videos and do not provide a stable proxy.

## 6 Conclusion

This work has presented MEMO for talking video generation. MEMO effectively alleviates temporal error accumulation and enhances long-term identity consistency via a new memory-guided temporal module, while generating videos with high lip-audio and expression-audio alignment via a new emotion-aware audio module. There are several promising directions remaining as future work below.

**Future work:** (1) Integrating **Diffusion Transformer** backbones [39] could enhance generation quality and expressiveness of talking videos. (2) Incorporating a dedicated head motion conditioning mechanism could enable **more dynamic head motion**. (3) Extending MEMO to broader applications, such as talking body generation, could unlock new possibilities for human video generation.

### Ethics Statement

We acknowledge that audio-driven talking video generation has strong positive potential in education, accessibility, virtual assistants, and entertainment, but also carries serious misuse risks. The same capability can be abused for impersonation, non-consensual synthetic media, fraud, harassment, and political misinformation. These risks are especially severe when real identities, voices, or sensitive contexts are involved. We restrict the release of our model and code to research use and explicitly prohibit malicious, misleading, defamatory, or privacy-infringing applications. We require users to obtain proper authorization for all inputs (*e.g.*, reference images and audio) and to comply with applicable laws, personality rights, and copyright regulations. For downstream deployment, we strongly recommend adding provenance signals (*e.g.*, watermarking/content credentials), maintaining auditable usage logs, and including human review before public dissemination of generated videos. Safeguards reduce but do not eliminate misuse. In addition, model behavior may vary across demographics, languages, and acoustic conditions due to data bias. We therefore encourage careful risk assessment, transparent disclosure of synthetic content, and continued research on detection and responsible release practices.

### Acknowledgments

This research is supported by the Ministry of Education, Singapore, under its MOE AcRF Tier 2 Award MOE-T2EP20223-0003. Any opinions, findings and conclusions or recommendations expressed in this material are those of the author(s) and do not reflect the views of the Ministry of Education, Singapore. In addition, this research is supported in part by NUS Start-up Grant A-0010106-00-00 and by the National Natural Science Foundation of China under Grant No. 62320106007.

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

# A    More Qualitative Results

**Comparisons with baselines.** Figure 15 showcases comparisons between talking videos generated by MEMO and baseline models on sampled out-of-distribution (OOD) data. Specifically, some baseline models (*e.g.*, Hallo [65] and Hallo2 [12]) tend to produce artifacts and fail to preserve the original identity and fine details. While certain methods (*e.g.*, AniPortrait [62] and V-Express [60]) generate videos with fewer artifacts, they suffer from poor lip-audio synchronization and motion smoothness. In contrast, our method demonstrates the ability to produce more natural facial expressions and head movements that are well-aligned with the audio input. Additionally, the videos generated by MEMO exhibit superior overall visual quality and stronger identity consistency.

**More visualization of emotion-guided generation.** The facial expressions of the generated talking video are influenced by both the expressions in the reference image and the emotional tone of the audio. As discussed in Section 4.2, the overall emotional tone of facial expressions is inferred mainly from the facial expression of the reference image, while our audio emotion-aware module functions mainly as a subtle adjustment to enhance or moderately alter the emotion when prompted by the audio. Figure 14 in Section 5.4 has demonstrated that given a fixed reference image, MEMO can refine the facial expressions of talking videos based on the given audio emotion.

In this appendix, we further explore the flexibility of our method by evaluating its ability to generate expressive talking videos using reference images depicting the same person with different emotional expressions, such as neutral, angry, happy, and sad. To isolate the effect of reference image expressions, we set the audio emotion label to match the emotional state of each reference expression. As shown in Figure 16, our method adapts seamlessly to diverse emotional states, generating highly expressive and emotionally consistent talking videos. These results highlight the robustness and versatility of MEMO in leveraging both reference expressions and audio cues to create emotionally nuanced talking videos.

# B    More Ablation Studies

**Classifier-free guidance scale.** By adjusting the classifier-free guidance scale, we observe variations in the expressiveness of the generated faces. As shown in Figure 17, higher guidance scales lead to more pronounced emotional expressions in talking videos.

**Dynamic motion frame training.** Appendix D presents a dynamic motion frame training strategy as a supplementary approach. The ablation results of this strategy, evaluated through human preference studies, are shown in Figure 18. Specifically, our method incorporating this strategy consistently receives higher human preference scores, highlighting its effectiveness in enhancing overall quality, motion smoothness, expression-audio alignment, lip-audio synchronization, and identity consistency.

**Inference-time runtime and memory.** Table 4 compares inference latency and GPU memory under the same hardware and inference settings. The memory mechanism adds a modest runtime overhead (+10.0 ms/step, about +2.5% compared with our variant without memory) while improving long-term consistency. It also introduces additional GPU memory for persistent memory states (about +3.2 GB peak in this setup). Importantly, this overhead is bounded with respect to generated video length because the memory uses fixed-size states with dynamic updates.

Table 4: Inference-time runtime and GPU memory comparisons.

| Method | Avg step time (ms) | Avg allocated (GB) | Peak allocated (GB) |
|---|---|---|---|
| Hallo | 327.1 | 5.5 | 8.0 |
| Ours (w/o memory) | 404.0 | 5.8 | 8.2 |
| Ours (w/ memory) | 414.0 | 9.0 | 11.5 |

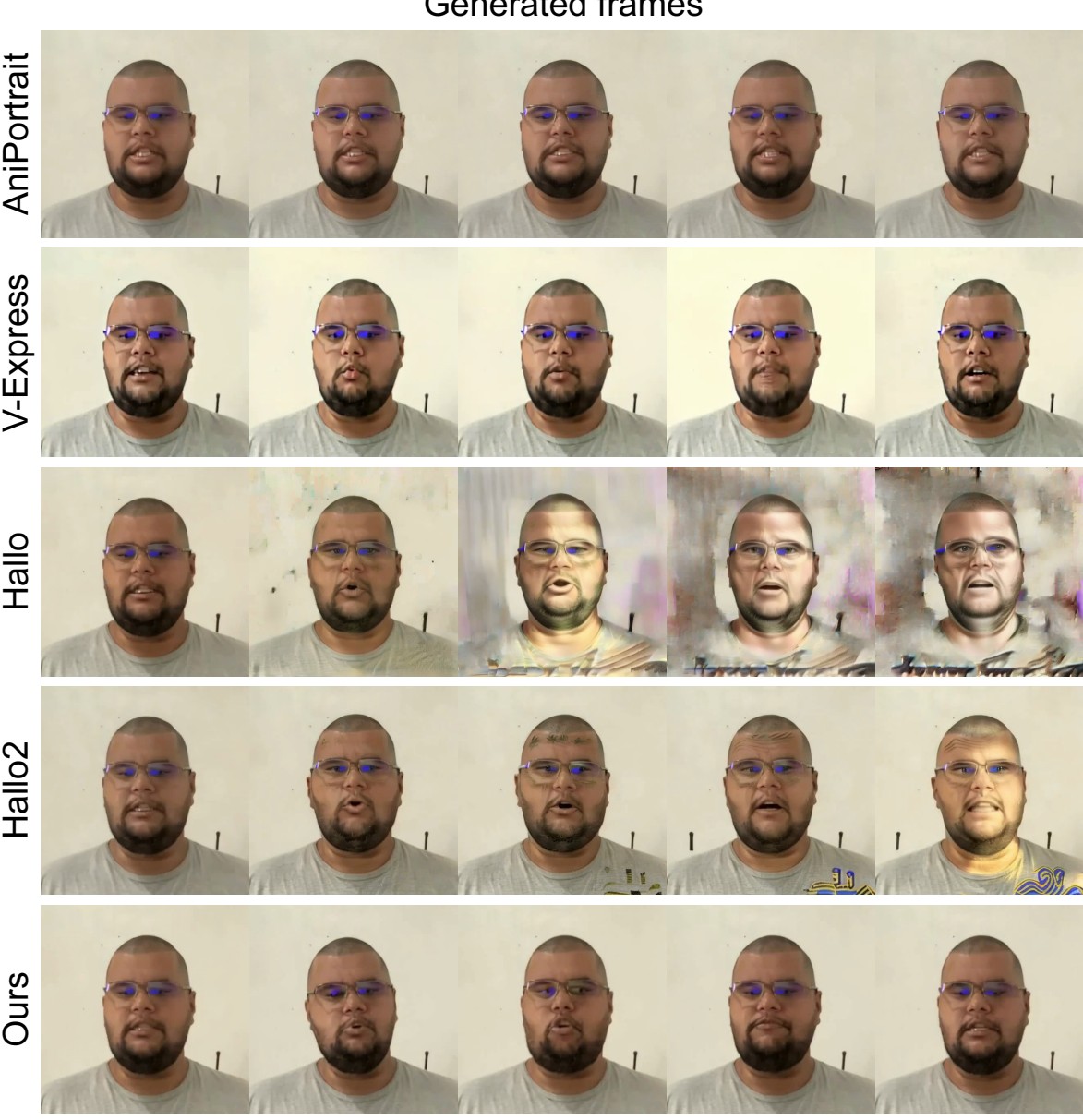

Figure 15: Visualization of generated videos on the OOD dataset. Existing methods either have poor lip-audio synchronization (*e.g.*, AniPortrait [62]) or suffer from error accumulation (*e.g.*, Hallo [65]). In contrast, MEMO generates talking videos with natural head motion and accurate lip-audio synchronization without artifacts. Please refer to the supplementary for video demos.

## C  Further Related Work

Diffusion models [47; 22; 69; 71] are highly expressive generative models, demonstrating remarkable capabilities in image synthesis [45; 41] and video generation [17; 64; 72; 19]. Stable Diffusion [45] employs a U-Net architecture and generates high-resolution images in the latent space, which is extended to video domains by AnimateDiff [17] by adding temporal attention layers. These models generate images or videos based on text prompts, where the text guidance from the pre-trained text encoder is introduced through cross attention. In the domain of talking video generation, diffusion models also show promising results in generation quality [18; 57; 62; 65; 50; 66], outperforming previous GAN-based methods [42; 76]. Instead of using text

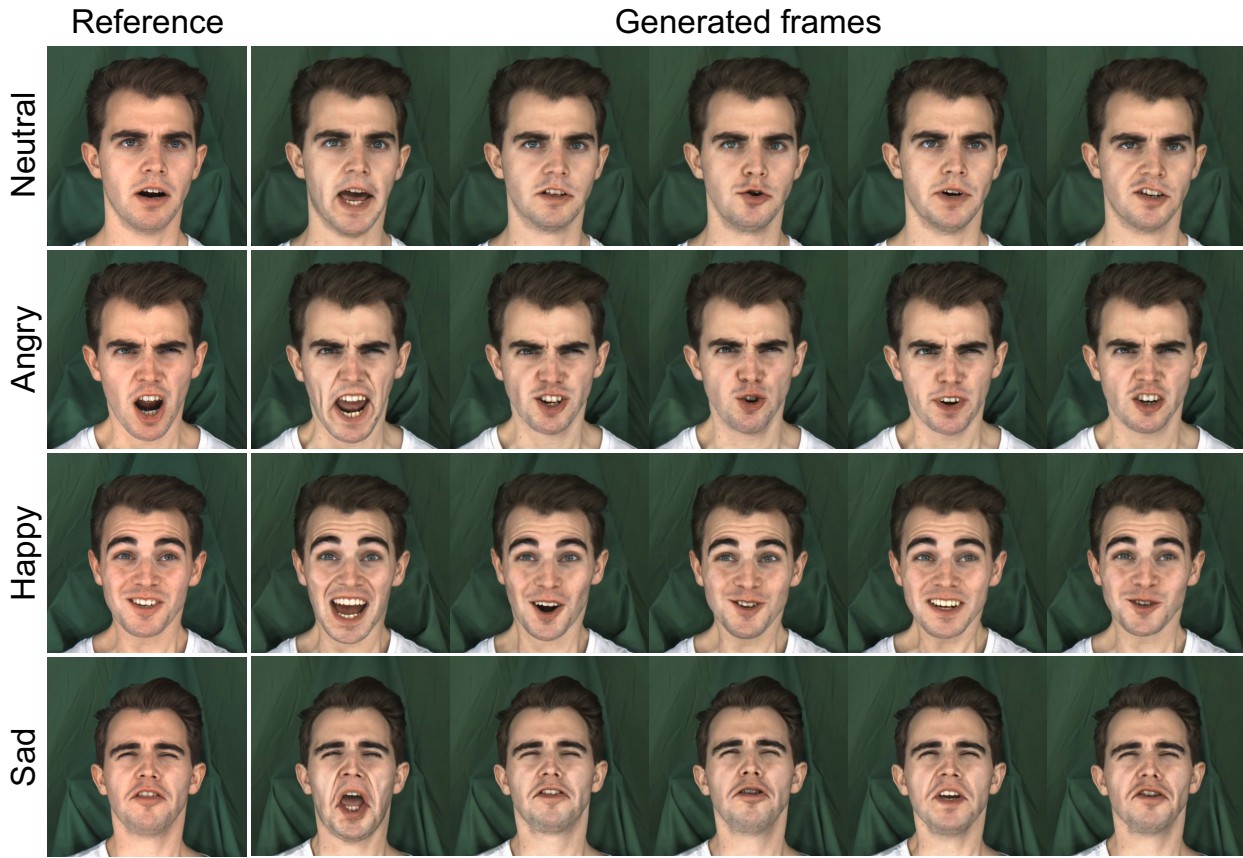

Figure 16: More visualization of expressive talking videos generated by MEMO based on reference images with various emotions. Please refer to the supplementary for video demos.

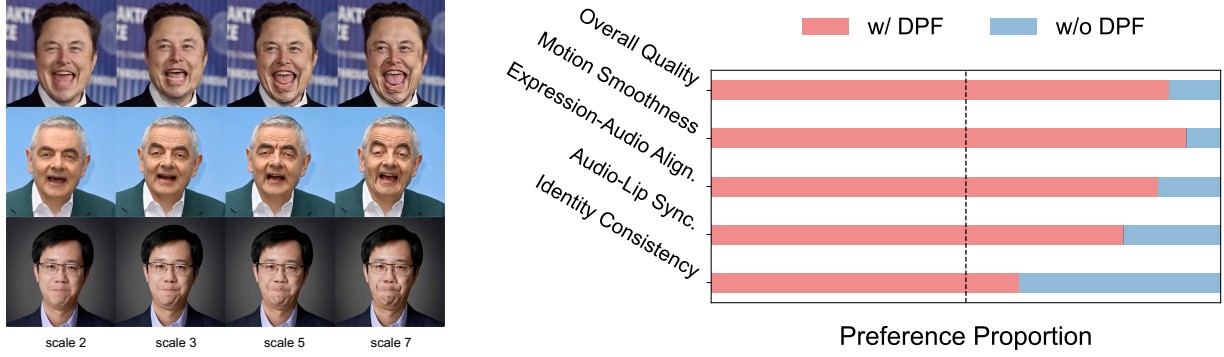

Figure 17: Ablation of the classifier-free guidance scale.

Figure 18: Human preferences between MEMO variants with and without dynamic motion frames (DPF) training.

prompts, most of these diffusion-based methods condition diffusion models on image and audio embeddings extracted from a pre-trained image encoder and audio encoder, respectively.

A recent work, MCDM [46], also proposes a motion-prior module to mitigate error accumulation in long talking video generation. While MCDM also improves temporal modeling abilities, its memory mechanism mainly focuses on motion priors and is comparatively less effective in ensuring cross-frame consistency and long-term identity preservation. Compared to MCDM, MEMO not only achieves more expressive and emotionally aligned facial animations through the emotion-aware audio module, but also addresses long-

term drifting via memory-guided linear attention. This allows MEMO to maintain identity consistency and smooth motion across long video sequences.

## D  More Implementation Details

Both Reference Net and the spatial module of Diffusion Net are initialized with the weights of SD 1.5 [45]. The temporal module of Diffusion Net is initialized from AnimateDiff [17]. Here, the Reference Net provides identity information from reference images for spatial modeling of the Diffusion Net and offers temporal information from motion frames for temporal modeling of the Diffusion Net. For both Reference Net and Diffusion Net, we replace the text cross-attention with image cross-attention. We add two projection modules to convert the audio embedding and image embedding into the dimensions required by our attention module. The audio embedding consists of all the hidden states from the Wav2Vec 2.0 model [2].

The training videos are center-cropped and resized to a resolution of $512 \times 512$ pixels. With a fixed learning rate of 1e-5, we train MEMO for 15k and 600k steps at training stages 1 and 2, respectively. During Stage 2, a fixed number of 16 motion frames is used to compute memory states as motion context (cf. Eq. 3 and Eq. 4). Specifically,

$$M_{KV}^{16} \leftarrow \sum_{j=1}^{16} \gamma^j \phi(K_{h,j}) V_{h,j}^{\top},$$
$$M_{K}^{16} \leftarrow \sum_{j=1}^{16} \gamma^j \phi(K_{h,j}),$$

where the memory decay factor $\gamma$ is set to 0.9. Besides 16, the number of motion frames can be dynamically chosen from 16, 32, or 48 for training, enabling the model to handle longer context scenarios more effectively. Nevertheless, thanks to the memory update mechanism with causal history decay (cf. Section 4.1), such dynamic motion frame training is unnecessary. Moreover, emotion embeddings, reference images, audio embeddings, and motion frames are randomly dropped with a probability of 5% for classifier-free inference. At inference, we set the frame rate to 30 frames per second (FPS) and employ autoregressive generation, producing 16 frames per iteration. The CFG scale is set to 3.5. When generating each sub-clip, the memory only needs to be updated once during the first denoise step, and subsequent denoise steps will not incur additional computational overhead. More details on the ablation of the CFG scale, dynamic motion frame training, and $\gamma$ can be found in Appendix B. Our model is trained on NVIDIA A800 GPUs.

## E  Audio Emotion Detection

To improve the natural expression of talking videos, we develop an emotion detection model to detect emotion labels from audio. In this appendix, we first introduce the data collection and processing strategies for audio emotion recognition, followed by the details of our emotion detector.

### E.1  Dataset Collection and Processing

**Dataset collection.** To achieve robust emotion detection across both speech and music audio sources, we collect a large-scale dataset encompassing both speech and music segments, each annotated with emotion labels. A detailed overview of the datasets used in our training process is provided in Table 5. For speech audio, we collect data from a recent Speech Emotion Recognition benchmark, EmoBox [34], which incorporates 23 datasets from various origins, covering 12 distinct languages. Regarding music audio, we gather data from the RAVDESS-song [33] and MTG-Jamendo [4] datasets, including songs with and without background music.

All data underwent a standardized processing protocol, converted to a monophonic format with a sampling rate of 16,000 Hz. Each utterance is uniquely annotated with an emotion label. For datasets containing lengthy samples, such as MTG-Jamendo, we divide them into shorter segments of 30 seconds to align with the typically shorter length of other datasets, assigning the same label to all segments. Each dataset was then split into training and testing sets with a ratio of 3:1.

Table 5: Statistics of the emotion detection Dataset. The source column represents the origin of the samples, and the language column specifies the dataset's language. #Emo indicates the number of emotion categories. #Utts shows the total number of utterances. #Hrs represents the total hours of training data.

| Speech Emotion Recognition datasets | | | | | |
|---|---|---|---|---|---|
| **Dataset** | **Source** | **Language** | **#Emo** | **#Utts** | **#Hrs** |
| AESDD [59] | Act | Greek | 5 | 604 | 0.7 |
| ASED [44] | Act | Amharic | 5 | 2,474 | 2.1 |
| ASVP-ESD [29] | Media | Mix | 12 | 13,964 | 18.0 |
| CaFE [16] | Act | French | 7 | 936 | 1.2 |
| EMNS [38] | Act | English | 8 | 1,181 | 1.9 |
| EmoDB [5] | Act | German | 7 | 535 | 0.4 |
| EmoV-DB [1] | Act | English | 5 | 6,887 | 9.5 |
| Emozionalmente [7] | Act | Italian | 7 | 6,902 | 6.3 |
| eNTERFACE [36] | Act | English | 6 | 1,263 | 1.1 |
| ESD [75] | Act | Mix | 5 | 35,000 | 29.1 |
| JL-Corpus [24] | Act | English | 5 | 2,400 | 1.4 |
| M3ED [74] | TV | Mandarin | 7 | 24,437 | 9.8 |
| MEAD [61] | Act | English | 8 | 31,729 | 37.3 |
| MESD [14] | Act | Spanish | 6 | 862 | 0.2 |
| Oreau [28] | Act | French | 7 | 434 | 0.3 |
| PAVOQUE [49] | Act | German | 5 | 7,334 | 12.2 |
| Polish [26] | Act | Polish | 3 | 450 | 0.1 |
| RAVDESS [33] | Act | English | 8 | 1,440 | 1.5 |
| SAVEE [23] | Act | English | 7 | 480 | 0.5 |
| SUBESCO [52] | Act | Bangla | 7 | 7,000 | 7.8 |
| TESS [13] | Act | English | 7 | 2,800 | 1.6 |
| TurEV-DB [6] | Act | Turkish | 4 | 1,735 | 0.5 |
| URDU [30] | Talk show | Urdu | 4 | 400 | 0.3 |
| Music Emotion Recognition Datasets | | | | | |
| **Dataset** | **Source** | **Lang** | **Emo** | **#Utts** | **#Hrs** |
| RAVDESS-Song [33] | Act | English | 6 | 1,012 | 1.31 |
| MTG-Jamendo [4] | Media | Mix | 56 | 5,022 | 299.47 |

**Label merging.** A major challenge in integrating different datasets is aligning their label spaces, as each dataset often features distinct emotion categories. For instance, the URDU dataset [30] contains only four emotion labels: happy, sad, angry, and neutral. In contrast, the ASVP-ESD dataset [29] includes 12 emotion labels, covering less common emotions such as boredom and pain. For music emotion recognition datasets like MTG-Jamendo [4], there are 56 mood/theme tags, not all of which correspond to emotional labels, and each sample can be assigned multiple tags. These discrepancies and overlaps in category spaces across different datasets present significant challenges for emotion detection.

To establish a generalized label space, we design our module to perform an 8-class classification task, selecting labels that are both commonly recognized and easily distinguishable: `angry`, `disgusted`, `fearful`, `happy`, `neutral`, `sad`, `surprised`, and `others`. We meticulously review and map the original labels from each dataset to fit within this new label space. For instance, samples labeled as `pleasure` in the ASVPESD dataset are mapped to the `happy` category due to their semantic similarity. Labels that do not clearly correspond to a specific emotion are categorized under the `others` label.

Table 6: Accuracy comparison of audio emotion detection between Emotion2vec [35] and our learned emotion detector.

| Dataset | Emotion2vec | Ours |
|---|---|---|
| AESDD | 75.84 | 78.52 |
| ASED | 86.20 | 85.23 |
| ASVP-ESD | 52.55 | 55.99 |
| CaFE | 73.30 | 100.00 |
| EMNS | 57.98 | 61.87 |
| EmoDB | 88.41 | 100.0 |
| EmoV-DB | 77.84 | 91.22 |
| Emozionalmente | 66.61 | 71.02 |
| eNTERFACE | 28.21 | 32.05 |
| ESD | 94.83 | 99.94 |
| JL-Corpus | 71.92 | 100.00 |
| M3ED | 42.59 | 41.52 |
| MEAD | 61.74 | 71.45 |
| MESD | 40.65 | 41.12 |
| Oreau | 50.96 | 42.31 |
| PAVOQUE | 85.15 | 92.74 |
| Polish | 44.89 | 100.00 |
| RAVDESS | 82.36 | 100.00 |
| SAVEE | 83.33 | 100.00 |
| SUBESCO | 78.43 | 100.00 |
| TESS | 76.29 | 95.14 |
| TurEV-DB | 47.45 | 53.47 |
| URDU | 54.00 | 56.00 |
| RAVDESS-Song | 43.58 | 100.00 |
| MTG-Jamendo | 65.30 | 74.50 |
| **Total** | **68.78** | **78.26** |

### E.2   Audio Emotion Detector

We train an 8-way classifier for our task, drawing inspiration from state-of-the-art methods in speech and music emotion detection. Our solution builds upon Emotion2vec [35], a robust universal speech emotion representation model. The feature extractor employs multiple convolutional layers and transformer blocks. It is trained using a teacher-student online distillation self-supervised learning approach. The feature extractor backbone of Emotion2vec is pre-trained on a large-scale multilingual speech corpus. For our classification task, we use the fixed Emotion2vec backbone as the feature extractor and train a 5-layer MLP as the classification head. To stabilize the training process, we apply gradient clipping, constraining the gradient updates within an $l_2$ norm of 1.0. To enhance the model's generalization ability, we incorporate a contrastive learning technique [70].

The test accuracy of our trained emotion detector for each dataset, as well as the overall accuracy, is reported in Table 6. Specifically, our trained detector achieves better performance than the original Emotion2vec [35], which adopted a single linear layer after the feature extraction backbone for downstream emotion detection.

### E.3   Robustness of Audio Emotion Detection

One possible concern is that the potential inaccuracy in audio emotion detection may lead to unreliable emotion labels. As shown in Table 6, although our emotion detector is not perfect, it has already surpassed existing open-source models in accuracy. This facilitates more reliable emotion detection and dynamic emotion adjustments in talking video generation. Moreover, as detailed in Section 4.2, to enhance the

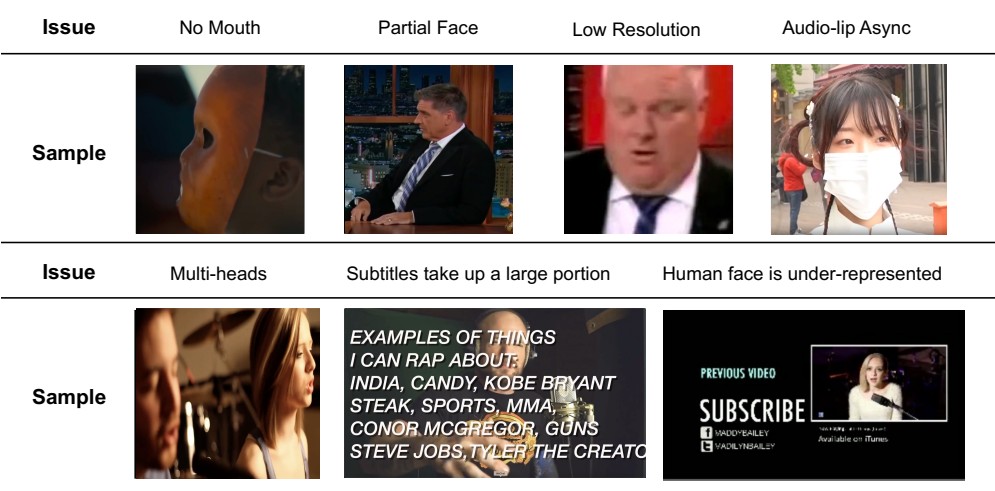

| Issue | No Mouth | Partial Face | Low Resolution | Audio-lip Async |
|---|---|---|---|---|
| **Sample** | | | | |

| Issue | Multi-heads | Subtitles take up a large portion | Human face is under-represented |
|---|---|---|---|
| **Sample** | | | |

Figure 19: Examples of issues in the raw dataset.

robustness of emotion labels, our emotion detection is performed at the segment level. Specifically, each segment's emotion is determined by the most frequently detected emotion across its frames, where each frame's emotion is evaluated using audio features extracted from a 3-second sliding window centered around that frame. This strengthens the reliability of detected emotion labels.

### E.4   Robustness of Audio Emotion Detection

One possible concern is that the potential inaccuracy in audio emotion detection may lead to unreliable emotion labels. As shown in Table 6, although our emotion detector is not perfect, it has already surpassed existing open-source models in accuracy. This facilitates more reliable emotion detection and dynamic emotion adjustments in talking video generation. Moreover, as detailed in Section 4.2, to enhance the robustness of emotion labels, our emotion detection is performed at the segment level. Specifically, each segment's emotion is determined by the most frequently detected emotion across its frames, where each frame's emotion is evaluated using audio features extracted from a 3-second sliding window centered around that frame. This strengthens the reliability of detected emotion labels.

More importantly, as stated in Section 4.2, the emotional tone of the generated video is primarily driven by the facial expression in the reference image, with audio-derived emotion labels providing subtle refinements. This further enhances model robustness, ensuring emotional coherence in generated videos even with noisy labels. Video demonstrations provided in the supplementary material further validate the effectiveness of our method.

## F   Data Processing Pipeline

We collect a comprehensive set of open-source datasets, such as HDTF [73], VFHQ [63], CelebV-HQ [77], MultiTalk [54], and MEAD [61], along with additional data we collected ourselves. The total duration of these raw videos exceeds 2,200 hours. However, as illustrated in Figure 19, we find that the overall quality of the data is poor, with numerous issues such as lip-audio misalignment, missing heads, multiple heads, occluded faces by subtitles, extremely small face regions, and low resolution. Directly using these data for model training results in unstable training, poor convergence, and terrible generation quality.

To further obtain high-quality talking video data, we developed a dedicated data processing pipeline for talking head generation. The pipeline consists of five steps:

• First, we perform scene transition detection based on TransNet V2 [48] and trim video clips to a length of less than 30 seconds.

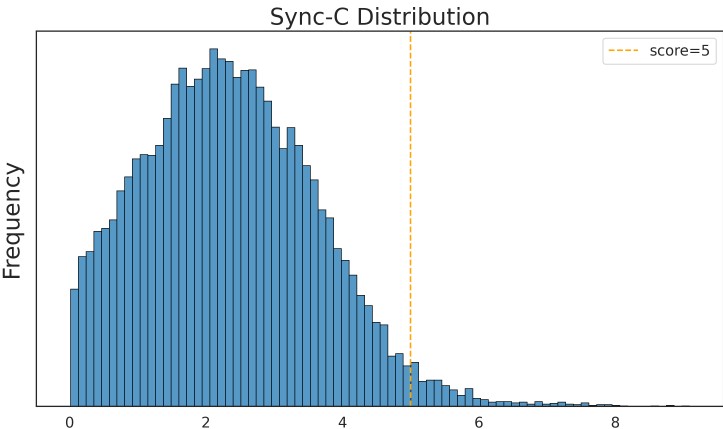

Figure 20: Distribution of the Sync-C in the CelebV-HQ.

• Second, we apply face detection based on Grounding DINO [31], filtering out videos with no faces, partial faces, or multiple heads, and use the resulting bounding boxes to extract talking heads. To ensure that the cropped areas encompass more than just the human faces, we apply a scaling factor 1.1 to the bounding box regions.

• Third, we use HyperIQA [51], an image quality assessment model, to filter out low-quality and low-resolution videos. We apply HyperIQA to the first frame of each video and find that when the IQA score exceeds 40, there is a noticeable improvement in overall video quality. Therefore, we use an IQA score of 40 as a selection threshold, but this threshold can be dynamically adjusted depending on the dataset quality requirements.

• Fourth, we utilize SyncNet [42] to filter out videos with lip-audio synchronization issues. The SyncNet Confidence (Sync-C) metric is used as the basis for filtering. Figure 20 illustrates the confidence distribution of the CelebV-HQ dataset, where a threshold of 5 is applied for filtering. This threshold, like others, can be dynamically adjusted based on the dataset quality requirements.

• Lastly, we manually check the lip-audio synchronization and overall video quality for more accurate filtering for a subset of the data. After completing the entire pipeline, the total duration of the processed high-quality videos is about 660 hours.

