# OpenReview forum: "MEMO: Memory-Guided Diffusion for Expressive Talking Video Generation"
_TMLR — Accepted by TMLR_

### Review · Reviewer_bbTE · 2025-11-01

**Summary Of Contributions:**

This paper proposes MEMO for expressive talking video generation that introduces a motion memory module into a latent diffusion model. Unlike prior talking-head approaches that generate frames sequentially without temporal feedback, MEMO maintains motion memory matrices that record key representations of the past frames. These are used to guide the denoising process for future frames, improving temporal consistency and expressiveness.

**Audience:**

Yes

**Audience Explanation:**

TMLR’s community includes researchers in topics included in this paper such as multimodal generation, speech-driven animation, and diffusion-based video synthesis.

**Claims And Evidence:**

Yes

**Claims Explanation:**

1. The paper provides clear architectural definitions and an explicit formulation of motion memory matrices, not just a conceptual claim.
2. Quantitative results show notable improvements in almost all metrics compared to previous baselines.
3. Ablations demonstrate that removing motion memory causes visible temporal degradation, confirming its contribution.
4. Visual results are consistent with quantitative metrics, where videos show smoother transitions and more expressive detail.
5. Overall, this is a very good work.

**Requested Changes:**

1. The paper would be significantly strengthened by a more detailed analysis and/or visualization of the motion memory mechanism. While Section 3.2 defines the memory as two matrices that record the past f frames, the paper does not illustrate what specific temporal or expressive information these matrices capture during generation. A qualitative or quantitative visualization, such as plotting the attention/related matrix module weights over time, or showing how memory activations evolve through the diffusion steps, would make the mechanism’s contribution clearer.
2. How does the memory mechanism affect training and inference cost? A brief analysis of runtime and memory usage would be helpful.

---

### Review · Reviewer_koxi · 2025-11-25

**Summary Of Contributions:**

This paper works on audio-driven talking video generation. The authors argued previous models mostly use a fixed context window and only conditioned on fixed emotion labels. The authors suggested to use a memory guided temporal module and a emotion aware audio model to ensure the consistency on long rollouts and better emotion conditioning. The memory guidance is achieved by encoding the reference image and motion frames with a reference net as the input for diffusion net. The audio condition is achieved with a pretrained wav2vec model and newly trained emption detector. The proposed method achieved better quantitive results compared to previous state of the art and could alleviated error accumulation in long-duration video.

Strength:
- The model will be open-sourced to boost research of related community.
- The overall design makes sense and achieved better results with reasonable increase in the complexity of model architecture.

Weakness:
- The contribution on overall methodology is incremental as the proposed method is a close follow-up of previous literatures with limited changes to the input modality and model architecture.
- The main paper does not include quantitive ablative study on the effectiveness of each design module. Quantitive metrics will help readers better understand the necessity of each module for future model design choices.

In conclusion, the paper provides valid insights of audio-driven video generation. The author also promised to open-source their checkpoints and source code which would be beneficial for future research. However, the contribution is limited and might have limited impacts to other research community as the use of multi-modality conditioning and memory guidance have been abundant in related fields.

**Audience:**

Yes

**Audience Explanation:**

Yes, the improvements on long video generation and audio conditioning could be beneficial to folks working on audio-driven video generation.

**Broader Impact Concerns:**

The video generative models with human subjects are a sensitive fields with regard to privacy and legal concerns. The authors have included an ethics statements but should still act with caution.

**Claims And Evidence:**

Yes

**Claims Explanation:**

Yes, the authors provided quantitive and qualitative evaluation of the proposed method in comparison of the previous methods.

**Requested Changes:**

Ablative study on the effectiveness of each design module. Quantitive metrics will help readers better understand the necessity of each module for future model design choices.

---

### Review · Reviewer_sG8R · 2025-12-18

**Summary Of Contributions:**

This paper presents MEMO, a diffusion-based talking-face generation system with two key components: 1. a memory-guided temporal module using linear attention over a compact memory state to improve long-term identity consistency and motion smoothness in long videos; 2. an emotion-aware multi-modal module, which replaces standard cross-attention with audio–visual multi-modal attention and modulates the UNet via emotion-adaptive LayerNorm using emotions detected from audio (or optionally user-specified labels).

The authors also construct a large, cleaned multi-dataset corpus and evaluate on VoxCeleb2 and two OOD benchmarks with both automatic metrics and human studies.

**Audience:**

Yes

**Audience Explanation:**

A diffusion-based talking-face generation system is definitely of interest for many individuals in TMLR's audience.

**Broader Impact Concerns:**

This method definitely raises some concerns on broader impact. In particular, regarding Deepfakes / impersonation / non-consensual media. A system that can generate highly realistic, emotion-controlled talking faces is inherently dual-use. It can be used for benign applications but also for impersonating real people, producing non-consensual or defamatory content, or targeted political persuasion. The paper should explicitly acknowledge these risks and discuss potential safeguards (watermarking, detection tools, usage restrictions, consent requirements).

**Claims And Evidence:**

Yes

**Claims Explanation:**

The paper claims that MEMO improves overall video quality, lip–audio synchronization, identity consistency, and expressiveness over prior talking-face methods, and Table 1 shows gains in FVD, FID, Sync-C/D, identity similarity.

The architectural claims about the two key modules are supported by mathematical descriptions and a detailed training strategy.

The claim that longer temporal context mitigates error accumulation and improves long-term consistency is supported by qualitative long-video comparisons (Figure 4) and a human study varying memory length (Figure 10).

However, there are also some claims that are only partially backed by evidence: the paper does not provide module-wise quantitative ablations on standard metrics (e.g., FVD / Sync-C with and without the memory or emotion modules), and there is no automatic classifier-based metric for emotion consistency.

Baselines are not retrained on the same cleaned dataset, which slightly weakens the fairness of the “consistently outperforms” claim.

**Requested Changes:**

There are some changes that would be a nice addition to the paper.

- Add clearer module-wise ablations
- Clarify fairness of comparisons: be explicit about the training data used for MEMO vs each baseline and briefly discuss how much of the gain might come from better data rather than only the architecture. Retraining one baseline on the same data would be ideal, but even just a discussion would be a plus.
- Include a light compute / efficiency summary

---

### Decision · Action_Editor_zAew · 2026-02-07

**Recommendation:** Accept with minor revision

**Additional Comments:**

The paper is recommended for acceptance subject to minor revisions.

Please incorporate into the final manuscript the clarifications and additional results already provided during the rebuttal, in particular:
- The module-wise quantitative ablations for the memory and emotion components;
- The inference-time runtime and GPU memory comparison table;
- A brief discussion clarifying the role of training data versus architectural contributions;
- An explicit discussion of broader-impact considerations and potential safeguards against misuse.

**Audience:**

Yes

**Audience Explanation:**

The work is clearly relevant to a substantial segment of the TMLR audience, particularly researchers interested in diffusion models, multimodal generation, speech-driven animation, and long-horizon temporal modeling.

The proposed memory-guided design addresses a well-known challenge in sequential generative modeling (temporal drift over long rollouts), and the emotion-aware conditioning mechanism is directly relevant to broader questions of controllability and expressiveness in generative systems. These aspects make the paper of clear interest to the TMLR community.

**Claims And Evidence:**

Yes

**Claims Explanation:**

The claims made in the submission are supported by appropriate evidence.

The paper provides clear architectural descriptions of the proposed memory-guided temporal module and emotion-aware audio module, along with extensive quantitative evaluations (e.g., FVD, Sync-C/D, identity similarity), qualitative comparisons, and human preference studies.

During the review process, the authors further strengthened the empirical support by adding controlled module-wise ablations under fixed training and evaluation conditions, which isolate the effects of the memory and emotion components. These results directly address concerns about error accumulation, long-term temporal consistency, and expression–audio alignment.

While some limitations remain (e.g., reliance on released checkpoints for baseline comparisons and limited automatic metrics for emotion consistency), these are transparently discussed and do not undermine the validity of the core claims.